# Comparative Analysis of the Effects of Daily Eating Habits and Physical Activity on Anthropometric Parameters in Elementary School Children in Latvia: Pach Study

**DOI:** 10.3390/nu12123818

**Published:** 2020-12-14

**Authors:** Ilze Justamente, Jelena Raudeniece, Liga Ozolina-Moll, Amelia Guadalupe-Grau, Dace Reihmane

**Affiliations:** 1Department of Human and Animal Physiology, Faculty of Biology, University of Latvia, Jelgavas Street 1, LV-1004 Riga, Latvia; ilze.justamente@lu.lv (I.J.); jelena.raudeniece@lu.lv (J.R.); liga.ozolina-molla@lu.lv (L.O.-M.); 2Department of Human Physiology and Biochemistry, Riga Stradiņš University, Dzirciema Street 16, LV-1007 Riga, Latvia; 3ImFINE Research Group, Department of Health and Human Performance, Universidad Politécnica de Madrid, 28040 Madrid, Spain; amelia.guadalupe@upm.es

**Keywords:** structured physical activity, eating habits, body mass index, pupils, elementary school

## Abstract

Growing incidence of obesity and related diseases in children poses new challenges and calls for a review of lifestyle habits. This study aimed to assess daily eating habits (EH) and physical activity (PA) levels and identify their association with obesity in 8–10-year-old children. Children’s EH and time spent in moderate-to-vigorous physical intensity (MVPA) was estimated from questionnaires (*N* = 1788). Weight, height, and waist circumference (WC) were collected, and body mass index (BMI) calculated. Girls consumed more fruits and vegetables, drank more water, and ate smaller portions of carbohydrate and protein rich foods but spent less time in MVPA compared to boys (*p* < 0.05). Obese children skipped breakfast more often and consumed less fruits and vegetables. Children who chose to eat in front of the screen had higher WC (62.88 ± 8.70 vs 60.59 ± 7.40 cm, *p* < 0.001) and higher BMI, and chose smaller vegetable portions and more calorie dense snacks (*p* < 0.001). 15.4% of pupils covered weekly MVPA recommendations with structured PA on weekdays. Increasing MVPA was related to a smaller number of unhealthy EH (*p* < 0.001). In conclusion, EH and PA levels differ between sexes and obese children have unhealthier EH. Higher levels of MVPA are related to healthier food choices, while pupils having meals in front of the screen have unhealthier EH and anthropometric measures. The majority of pupils did not reach the WHO recommendations of MVPA through structured PA on weekdays. Association between factors (EH and time spent in PA) and BMI was not found in this study.

## 1. Introduction

Overweight and obesity rates among children are rising in parallel with increased consumption of high energy dense foods, drinks, and inactive lifestyle causing health issues such as diabetes, cardiovascular and musculoskeletal diseases, and even cancer [1]. In addition, childhood obesity is more likely to be persistent into adulthood [2,3], also causing a variety of psychosocial complications [4]. Thus, the epidemic of obesity is raising concerns not only on the individual level, but also on the social level. Early interventions and lifestyle changes during childhood are the most important measures to prevent obesity-associated diseases in adults [5], where family lifestyle habits play a crucial role in the formation of the child’s behavior.

The World Obesity Federation predicts that by the year 2030, the number of obese children aged 5–19 will grow from 158 million in 2020 and reach 254 million [6]. In 2006, the World Health Organization initiated the European Child Obesity Surveillance Initiative (COSI) to monitor obesity in childhood and to use the data to introduce changes in overweight and obesity prevention programs. In 2018, results for Latvian pupils aged 9 showed that 27.2% boys and 22.9% girls were overweight or obese. 58.3% of these children participated in moderate-to-vigorous physical activity (MVPA) for at least 4 h weekly, but during the weekends 56.9% used an iPad, smart phone, computer and watched TV for 3 h or more [7]. Importantly, overweight and obesity have increased in comparison with 2008. 

Healthy eating patterns and PA are associated with healthy BMI among children. Unhealthy dietary patterns such as breakfast skipping, overeating, and rare participation in family meals are associated with total and central body adiposity, increased BMI and waist–to–height ratio [8,9]. To maintain healthy weight and avoid unhealthy weight gain, it is important to balance energy intake through healthy foods and beverages and energy expenditure with basal metabolic rate (BMR) and PA. Previous studies reported that children with higher BMI had unhealthier EH and more sedentary lifestyle habits [10]. For example, it has been shown that obese and overweight pupils more often skipped breakfast [11], consumed more ultra-processed calorie dense foods [12] and had lower MVPA levels [13,14]. 

According to EuroStat data, the obesity rate in adults in Latvia (in the population aged 18 years and over) is one of the highest in the European Union (following Malta, which is ranked first in obesity prevalence) [15], so it is particularly important to pay attention to children’s health indicators to reduce the risk of obesity and obesity-related diseases in the future. Improvements in diet and increase in PA have been shown to be cost-effective in terms of reducing medical costs, thus being an important part of various obesity prevention strategies [16,17].

There are some studies examining EH and PA in various age groups and sexes [7,18]; however, differences in behavioral patterns among weight groups have not been described in Latvian children. Therefore, this study aimed to analyze EH, PA, and anthropometric parameters among 8–10-year-old children in different weight groups and by sex, and to determine the relations and impact of measured parameters on BMI and WC.

## 2. Materials and Methods 

### 2.1. Study Design and Subjects

This paper describes the baseline characteristics of pupils enrolled in the Physical Activity and Children Overall Health (PACH) study and presents data related to this cross-sectional quantitative research design. The study involved 1788 8–10-year-old pupils from the second (*n* = 1119) and third class (*n* = 669) of which 53.5% (*n* = 957) were boys and 46.5% (*n* = 831) were girls who joined the Latvian Olympic Committee project “Sport for All Classes” (SFAC) in the school year 2019/2020. Project SFAC is supported by the Ministry of Education and Science of Latvia and ensures 5 vs 2 school-based PE lessons per week, promoting structured physical activities in schools. To participate in SFAC, school authorities (principals, form teachers, and physical education teachers) had to submit an official application with the list of pupils and corresponding signatures of their parents. 

The PACH study is an effective collaboration project between the University of Latvia and the Latvian Olympic Committee (ZD2019/20861) and has been developed in line with the Declaration of Helsinki (World Medical Association) and corresponds to ethical principles for medical research involving human subjects. 59 different schools representing all five statistical regions of Latvia, Vidzeme (13), Latgale (1), Zemgale (7), Kurzeme (11), Riga (27), were enrolled in the study through convenience sampling. The PACH study has been approved by the Ethical Committee of the Institute of Cardiology and Regenerative Medicine, University of Latvia (Nr.179/2019; effective from 14.10.2019). Parents were asked to sign a detailed written consent explaining all procedures and possible risks, and data collection and management procedures for participation in the PACH study providing that their child was taking part in the SFAC project. Verbal consent from subjects was obtained and they had the right at any point without explanation to refuse further participation in the study. 

### 2.2. Measures

#### 2.2.1. Anthropometric Measurements

To estimate obesity prevalence among 8–10-year-old pupils, data on weight, height, and WC were collected. Based on these measurements, BMI for each child (body weight (kg)/body height squared (m2)) and percentile was calculated [19]. After calculation of BMI, four study groups were distinguished: underweight (UW, <5th percentile); normal (NOR, 5th–85th percentile); overweight (OW, 85th–95th percentile); obese (OB, above 95th percentile). Anthropometric measurements were also compared between sexes.

#### 2.2.2. Eating Habits

To evaluate pupils’ EH among study groups and by sex, a questionnaire based on the content of the healthy eating guidelines of the Ministry of Health of the Republic of Latvia and a project of the United States Department of Agriculture “My Plate” was designed. It included an estimation of 7 EH commonly described in the scientific literature: eating breakfast [20], number of daily meals [21], fruit and vegetable consumption [22], choice of snacks [23] and beverages [4,24] and eating culture, e.g., watching TV while eating [25,26]. Current evidence suggests that eating while viewing the screen leads to obesity in children [27], thus an additional analysis estimating differences in BMI, WC, and other measured parameters were analyzed in pupils eating at the table and pupils eating in front of the screen. More detailed description of the survey items, answers, and correspondingly adapted scoring system (0–6 points) for the total number of unhealthy EH [11,20,21,24,28,29,30,31,32,33,34,35,36] is presented in Table A1.

#### 2.2.3. Meal Portion Size

Daily energy intake is considered one of the main factors causing obesity causing. Larger meal portions result in higher daily energy intake. To determine the dietary intake among different study groups and sexes, meal portion size for carbohydrate foods, protein rich foods and vegetables/salads were estimated. Pictures representing the size of portions were chosen from the Photographic Atlas of Food Portions for the Emirate of Abu Dhabi, 2014 [37]. More detailed description of the estimation of portion sizes in grams [32] is presented in Table A1.

#### 2.2.4. Time Spent in Structured Physical Activity

It has been shown that health-related physical fitness is primarily associated with PA during PE lessons and in recess, and engaging in sports/dance, but not with participation in nonorganized physical play outside school [38], thus the current study focused on structured PA on weekdays. Questions about such daily habits as means of getting to school, number of PE lessons in school, and participation in professional training (team and individual sports) and activities during breaks between school lessons were asked. Pupils’ answers were used to calculate time spent in light (<3 youth metabolic equivalents (METy), moderate (3–6 METy) and vigorous (>6 METy) PA as well as to estimate the prevalence of pupils reaching the WHO recommendations of MVPA per weekdays and for whole week. More detailed description of the survey items, answers, and correspondingly adapted METy from the Youth Compendium [39] is presented in Table A2.

#### 2.2.5. Total Energy Cost during Structured Physical Activity

Total energy costs were estimated based on METy from the Youth Compendium [39], a computed BMR, and duration of the specific activity, as follows: Total energy cost (kcal) = METy x BMR (kcal/min) × duration (min), where BMR for 3–10-year-old girls and boys is predicted from Schofield equation (BMR (kcal/min) = (20.315 × Weight (kg) + 485.9)/1440 and BMR (kcal/min) = [22.706 × Weight (kg) + 504.3]/1440, correspondingly). Study groups were described in relation to time spent in PA (reaching or not reaching WHO recommendations). In these calculations, *N* was comparatively lower, because children lacking data for BMI calculation and with incomplete/incorrect questionnaires about PA were excluded from the analysis. More detailed description of the estimation of total energy cost during structured PA is presented in Table A2. 

### 2.3. Procedure

Examination of the subjects occurred in the school setting. Research team visited pupils once between November 2019 and January 2020. Questionnaires estimating EH, meal portion size, and PA levels were electronic, and pupils filled them out on school computers with the assistance of their teacher and/or researcher. Additionally, questions were represented with pictures to keep children’s attention and facilitate the answering process. 

Medical nurses working in the schools were trained by the study research group and assisted in the collection of anthropometric measurements performed according to a standardized protocol (COSI) [40] and local school equipment was used. Body height was measured using a stadiometer, standing straight as possible and arms hanging freely along the sides. Height was measured in centimeters and the reading was taken until the last completed 1 mm (0.1 cm). The body weight of the children was measured in kilograms and recorded to the nearest 100 g (0.1 kg) unit. Children were weighed without shoes in light sportswear. Waist was measured to the nearest 0.5 cm with an anthropometric nonelastic measuring tape after normal expiration, at the level of the umbilicus. Sample size across variables varies due to data collection on different days (in some of the schools), sickness, or refusal to participate in particular measurements. Out of 1788 pupils recruited in the study, BMI was determined for 1645 children. 

### 2.4. Statistical Analysis

Shapiro-Wilks test was used to check the data for normality. Any differences between study groups are explained using the nonparametric (Mann-Witney U test) test. Quantitative data defined by more than two categories were tested using the Kruskal-Wallis test. The Chi Square test was used to analyze differences between study groups reported as categorical data. To see which categories have the largest difference between the expected counts, adjusted residuals (−2, 2) were used. Spearman’s correlation test was used to determine relations between variables. Characterizing the strength of the interaction between the traits, the correlation was evaluated as close (if *r* ≥ 0.7), medium (0.3 < *r* < 0.7), or weak (if *r* ≤ 0.3). Multiple linear regression (MLR) was used to model the linear relationship between the independent variables (number of daily meals and additional snacks, fruit and vegetable consumption (number of portions per day), meal portion sizes described in point 2.2.3., total time spent in PA and subgroups (light, moderate, vigorous) and dependent variables (BMI and WC)). Data were processed with SPSS 21 (Statistical Package for the Social Sciences, IBM, Armonk, NY, USA), expressed as median and quartile I (25%) and III (75%) and as mean rank. 

## 3. Results

### 3.1. Anthropometric Measurements

BMI prevalence among the pupils were 4.9% UW; 72.3% NOR; 9.4% OW and 13.3% OB. It was determined that 9% of boys were OW and 15,5% were OB, while 9.9% of girls were OW and 10.7% were OB. Obesity prevalence was found to be significantly higher in boys compared to girls (χ2(3) = 8.37, *p* = 0.039). Table 1 represents the median anthropometric data of the study groups and by sex.

### 3.2. Eating Habits

Table 2 summarizes the study data on six EHs assessed in 8–10-year-old pupils. Dietary surveys indicated that most of the pupils (59.2%) had their breakfast every day, 16.2% almost every day, 22.9% sometimes, while only 1.9% children admitted that they did not eat breakfast at all. Statistical analysis showed that obese children skipped breakfast more often than children in NOR or UW groups (χ2(9) = 8.76, *p* = 0.033, Table 2), while the difference between sexes was not observed. 

Number of daily meals was estimated as one of the EH. The majority of pupils (47.7%) reported having the recommended three meals per day followed by four and five meals (24.6%), two or fewer meals (24.3%), and six or more meals (3.6%) per day. Average number of daily meals was similar in all weight groups and did not differ between sexes (data not shown). 

Study data showed that 46.1% of pupils ate five–eight recommended fruit and vegetable portions per day. Girls reached the recommendations more often (χ2(1) = 5.70, *p* = 0.19, Table 2) while a significant difference between study groups was not found (Table 2). However, study data showed that OB pupils on average ate less fruit and vegetable portions per day in comparison to OW weight group (H(3) = 8.75, *p* = 0.033, Table 2). When viewed separately, girls chose to eat more portions of vegetables than boys (mean rank for girls 874 vs mean rank for boys 820, U = 332802.50, z = −2.37, *p* = 0.018). However, children in the OB group ate fewer portions of vegetables compared to NOR and OW groups (701 vs 787 vs 820 mean rank, respectively, H(3) = 8.53, *p* = 0.036), correspondingly, less often reaching the recommended three or more portions of vegetables per day (χ2(3) = 8.55, *p* = 0.036, Table 2). Similarly, 72,8% of pupils ate the recommended two or more portions of fruits per day, and OB pupils reached the norm less often (χ2(3) = 8.16, *p* = 0.043, Table 2). Study data showed that pupils independent of study group or sex preferred fruits more than vegetables (*N* = 415 choose more vegetables, *N* = 524 choose more fruits, Wilcoxon Signed Rank Test, T = 250529, z = −3.84, *p* < 0.001) in their daily menu.

Fruits (55.7%) were also the most often mentioned snack among all study participants, followed by nuts and dried fruits (11.7%), salty snacks (e.g., chips, fries) (10.7%), cookies and biscuits (8.6%), yogurt (8.5%) and chocolate, candies and ice cream (4.9%). Similarly, data showed that fruits (49.7%) were the most popular snack choice between meals for pupils (*n* = 754) that marked only one favorite snack. Snack choices in different weight groups were similar, but pupils in OW group tended to eat more dried fruits and nuts, while pupils in UW group tended to eat more biscuits, muffins, and cakes (χ2(15) = 25.07 *p* = 0.049, Figure 1). As can be seen in Table 2, a significant difference between the use of calorie dense snacks among study groups or sexes was not observed, however, boys tended to eat more calorie dense snacks than girls (χ2(1) = 2.91, *p* = 0.093). Data analysis also revealed that boys chose to eat salty calorie dense snacks more often, while girls chose such sweets as cookies and muffins (χ2(5) = 34.34, *p* < 0.001). In all study groups, the number of daily snacks taken was similar. Most of the children (48.7%) chose to eat two–three additional snacks per day, followed by 22.0% eating one, 17.7% eating four and five, while 8.5% of children ate six and more additional snacks per day. Differences between study groups and sex were not observed.

Less than half of the children (41.0%) chose water to quench their thirst, followed by tea (25%), fruit juices (17.3%), and sweetened drinks (16.3%) such as Coca Cola, Fanta, and Sprite. Girls drank water more often than boys, while boys drank more sweetened beverages (χ2(3) = 41.77, *p* < 0.001, Table 2). Significant differences between study groups were not observed (Table 2).

Most of the children indicated that they usually had a meal at the table alone or with the family (71.1%). However, as shown in Table 2, boys watched TV, looked at the computer, or used mobile devices during the meal more often than girls (χ2(1) = 14.85, *p* < 0.001). Similarly, OB and OW pupils tended to eat more in front of the screen in comparison to other study groups (χ2(3) = 7.10, *p* = 0.069, Table 2). Those children who chose to eat in front of the TV, computer or scrolling a phone had a higher BMI (U = 228563.50, z = −2.55, *p* = 0.011), and waist circumference (U = 205913.00, z = −4.02, *p* < 0.001) and unhealthier eating habits (Table 3).

In a sample size of 754 pupils, most of the children had two or less unhealthy EH, while 32.5% of the children reported three or more unhealthy EH like skipping breakfast, eating while watching TV, not eating enough vegetables, etc. Unhealthy EH were more common among boys compared to girls (1.94 ± 1.39 vs. 1.66 ± 1.45, *p* = 0.003, Table 2). 

### 3.3. Meal Portion Size

Boys in comparison to girls chose to put on the plate a bigger portion of carbohydrate containing foods such as potatoes, pasta, rice (U = 304295.00, z = −5.25, *p* < 0.001), as well as protein-rich foods such as meat and fish (U = 299890.00, z = −5.67, *p* < 0.001) (Table 4) while portion sizes were similar in all weight groups (data not shown).

### 3.4. Time Spent in Structured Physical Activity

The majority of the pupils reported that they use every opportunity to move actively, because they like to do so (44,6%), while less than 5% of children reported that they do not like to move actively at all, preferring to sit still or walk calmly. Children in the OB group reported that they prefer to be less active compared to UW, NOR, and OW groups (χ2(12) = 37.16, *p* < 0.001). 

More than half of the children (52.4%) got to school by car, 33.7% walked to school, and 13.9% used public transport. Data showed that most children lived relatively close to school and it took less than 10 min to get to school, both on foot and by car. Almost 70% of the study participants had an additional PA besides the PE at school. Furthermore, 80.4% of them did sports training 2–4 times a week. Data showed that boys were more likely to report additional PA outside the school curriculum and they had more training sessions per week compared to girls (3 (2;4) vs 2 (2;3)). More than half of the pupils (57.1%) preferred to be active in school breaks (running, playing active games), and again boys reported being more active compared to girls (χ2(4) = 63.02, *p* < 0.001). When asked what pupils prefer to do in their free time, one-third of children (33.8%) answered that they usually watch TV, play computer games, or use the telephone. Only a quarter (26%) of children chose to play with friends, sisters, or brothers. Less popular free time activities were reading, helping parents with household work, or going for a walk with a dog (14.4%, 12%, and 11.8% respectively). 

On average, in structured PA on schooldays (5 days a week), 49.8% of the pupils reached the WHO recommendation of 60 min of MVPA per day (300 min per 5 days), while only 15.4% reached this throughout the whole week (420 min in total, respectively). Study data showed that boys more often reached the WHO recommendations than girls, primarily due to increased time spent in MVPA (χ2(1) = 50.07, *p* < 0.001, Table 5). Only 22% boys and 17% girls reported participation in vigorous PA. Total time spent in structured PA on weekdays was 550 (375; 650) min, while in structured MVPA this was 295 (175; 390) min but was significantly higher in boys when compared to girls (U = 319095, z= −3.690, *p* < 0.001 and U = 286151.00, z = −6.98, *p* < 0.001, respectively). Differences between the weight groups were not observed (data not shown).

### 3.5. Total Energy Cost During Structured Physical Activity

The total energy costs to sustain previously described PA levels are listed in Table 6 according to whether pupils had or had not met the WHO recommended 60 min of MVPA during every weekday. There was a significant difference (*p* < 0.01) in the total energy costs between most of the groups except the ones noted in Table 6. 

### 3.6. Correlations Between Measured Parameters

Several correlations between parameters characterizing EH and PA and anthropometric measurements were found in the whole study population (Table 7). Correlation analysis of separate groups showed that time spent in PA in OB pupils were related to lower BMI (*r* = −0.141. *p* = 0.044) and lower WC (*r* = −0.182. *p* = 0.010). In addition, OB pupils who skipped breakfast more often had higher waist circumference (*r* = −0.163; *p* = 0.041).

Higher total energy costs were also related to bigger portions of carbohydrate-containing foods (*r* = 0.130, *p* < 0.001) and bigger portions of meat or fish (*r* = 0.166, *p* < 0.001). In addition, the relation between increasing structured PA (total time spent in light to vigorous PA) and lower number of unhealthy EH was found, *r* = −0.112, *p* = 0.002)

### 3.7. Multiple Linear Regression

A multiple linear regression model was used to assess whether time spent in structured PA and EH affected anthropometric measurements (BMI and WC) but gained no significant results. The results of the regression indicated that ten predictors explained only 1.1% of the variance and that the model was a significant predictor of WC (R2 = 0.011, F (9,1503) = 1.93 *p* = 0.045). It was found that the number of vegetable and fruit portions per day was a significant predictor for WC (*β* = −0.288, *p* = 0.014) and total time spent in light physical activity (*β* = −0.005, *p* = 0.042), whereas other factors did not show significant results. The results of the regression indicated that ten predictors explained only 0.5% of the variance and that the model was not a significant predictor of BMI (R2 = 0.005, F (9,1544) = 0.845 *p* = 0.575).

## 4. Discussion

Excess weight is an alarming problem worldwide not only among adults but also in the younger population which is more likely to become overweight or obese in adulthood, increasing the risk for the development of non-communicable diseases even in the period of adolescence [41]. The World Obesity Federation reported that in 2020 obesity reached a critical number of 158 million obese children, predicted to increase to 254 million by the year 2030. In 2016, there were 11.5% and 6.5% boys and girls aged 5–9 with obesity in Latvia [6]. Our study showed a prevalence of 13.3% of obese children, 15.5% of boys, and 10.7% of girls, respectively (significantly higher in boys). These growing numbers indicate the need for determination of unhealthy lifestyle habits among children and the development of effective methods for prevention of obesity at such a young age.

### 4.1. Unhealthy Eating Habits and Meal Portion Sizes

Unhealthy EH and low PA levels are the major reasons for increasing obesity levels and are quite common among children all over the world. Skipping breakfast in childhood tends to become a persistent habit in adult life promoting overweight and obesity [42].In Latvia the number of 11-year-old children eating breakfast regularly has decreased in a decade from 76.3% t0 67.4% in the year 2014 [18], while our study reported that only 59.2% pupils chose to have breakfast daily. In addition, the habit of having a regular breakfast tends to decrease in parallel with the increasing age of children [28]. In 2018, a study of children’s anthropometric parameters and school environment reported that 3.2% of 9-year-old Latvian children skipped breakfast always [7], which is similar to our findings, while this is two-three times lower in comparison to children in Britain [43] and Italy [28]. Our study showed that in the OB group, skipping breakfast was more common than in NOR and UW groups, supporting the hypothesis that irregular breakfast leads to increased body weight. The rapid lifestyle of the modern world could be the main reason for unhealthy EHs such as skipping breakfast [44], thus it is of importance to highlight the need to find the time for a proper breakfast so it becomes a tradition.

It is recommended to consume three meals daily and one–three healthy snacks per day [45]. In addition, it is important to balance nutrient intake for proper growth and development. The current study is in line with studies performed in Lithuania [11] and Poland [29] and showed that almost 50% of pupils have the recommended number of meals daily.

The WHO suggests consumption of more than 400 gr of fruits and vegetables per day (five portions)as a part of a healthy diet low in fat, sugar and salt, reducing the risk of obesity [46]. According to the literature, vegetable daily consumption in 7–15-year-old children is critically low—approximately 25%–30% [7,18,47]. In Finland 43.3% of 11-year-old children [21] and in Italy 48.8% of 8–9-year-old children [28] avoid eating fruit and vegetables daily. In the current study, we analyzed the frequency of fruit and vegetable portions per day as the number of portions can unfold the real (more precise) fruit and vegetable EH. The results showed that only 7.3% of children do not eat vegetables at all, but more than one-third of pupils eat the recommended three or more portions of vegetables per day. Contrary, more than two-thirds of pupils eat the recommended two or more portions of fruits daily, while 2.8% of children do not eat fruits. Our study data also revealed that OB pupils less often reach the norm for both fruit and vegetable intake. In line with our data, children and adolescents in Canada who consumed fruit and vegetables five or more times per day did not belong to the overweight or obese group [48]. Low intake of vegetables and fruits has been associated with increased risk of obesity [49] and a small inverse association between BMI and eating fruit and vegetables has been reported [50]. However, our study data showed no such correlation, which is in line with the results of an Health Behaviour in School-Aged Children study in 2005 [47]. A recent systematic review reported that girls tend to have a higher or more frequent intake of vegetables and fruit than boys [22]. The current study showed that girls consume more portions of vegetables per day than boys, but there is no difference in fruit consumption. Consumption of fruits and vegetables is still inadequately low (especially in OB pupils) as less than half of the pupils reach five or more portions per day. This may be explained by such factors as inappropriate opinions about fruit/vegetable prices, low income and lack of knowledge [51].

Fresh fruits and vegetables are considered as the best snack [52] while low nutrient and energy dense snacks impair quality of nutrition and lead to weight gain [53]. In the current study, approximately half of the pupils reported fruits as the most common snack, but less than one quarter of children chose unhealthy snacks, while other studies report that sweet snacks every day are consumed by 27.8% of 11–15-year-old children [7] and 38.1% of 7–9-year-old Latvian children [18]. It has been reported that boys eat more sweet snacks than girls [47]. Contrarily, the current study reported higher consumption of salty snacks in boys, and higher consumption of sweet baked good in girls. Furthermore, there was no association between snack choice and body measurements, except a weak correlation between WC and unhealthier snacks in OW group.

High consumption of sugar as sweetened beverages (SSBs) in young children is determined by (a) time spent watching TV; (b) parenting model; (c) school nutrition policies [54], and leads to weight gain and adverse health outcomes. Despite TV and online advertising, in the current study the most often chosen daily beverage, by 41% of children, was water, while one-third of children chose SSBs. In comparison, a study in similar age groups reported 6.5% of pupils using SSBs daily [7,18]. Daily SSBs consumption in different studies varies from 22% [29] to 40% [28] and more, and may provide an extra 350 kcal for each 750 mL of SSBs [55].

Nowadays, eating culture has changed significantly. Time for family meals at the table often is replaced with TV, computer, and other devices, which is related to higher intake of SSBs, chips and decreased intake of fruits [56]. Indeed, the current study data showed that almost one-third of children used their mobile devices or watched TV during their meal. These children had significantly higher WC and BMI and in fact they ate significantly smaller portions of vegetables and more often chose calorie dense snacks, reflected also in the higher number of unhealthy EHs. There is strong evidence available reporting increased consumption of fast food, SSBs and decreased consumption of fruits and vegetables while watching TV [26] for more than two hours per day [28]. The current study shows that obesity prevalence is higher in boys than girls. As boys use mobile devices during the meal more often than girls, this could be one of the explanations for these sex differences. Other aspects could be that boys chose to have a larger portion of carbohydrate rich food on their plate daily and the average number of unhealthy EH in boys was higher.

Large meal portions increase daily energy intake and are considered one of the main factors causing obesity [57]. The current study data showed that boys in comparison to girls chose to put on the plate a bigger portion of carbohydrate and protein rich containing foods. Contrary to the findings of a recent study which showed that large meal portions favor excess weight gain in early childhood [58], there were no differences between the weight groups in this study.

### 4.2. Physical Activity

Along with a healthy diet, PA can have a fundamental role, not only in a child’s growth period, but also in the prevention of non-communicable diseases throughout life [59]. However, a number of epidemiological studies have alerted concerns about low PA and high consumption of calorie dense foods in children [59]. Current data show that 8 out of 10 children enjoy being physically active, but pupils in the OB group preferred being less active compared to other weight groups (*p* < 0.05). Boys also reported having more additional PA outside the school, more training sessions per week, and were more active during breaks compared to girls, resulting in significantly higher time spent in MVPA and correspondingly increased total energy costs. However, only half of the pupils reached the WHO recommended 60 min of MVPA on weekdays during school time and additional training. If these structured PAs are the only ones throughout the week (with sedentary activities during weekends), then the number of children who reach the necessary amount of MVPA drops below 20%. In addition, pupils in this study participate in the SFAC project which ensures five PE lessons per week instead of the two determined by the Ministry of Education and Science of Latvia. This could mean that most of the pupils in Latvia on average may have 1.5–2 h of MVPA less than average participants in SFAC. Interesting, The Childhood Health, Activity, and Motor Performance School Study Denmark reported no differences in overall PA levels between children attending sports schools and normal schools, as children who were more active during school time were less active during leisure time [60]. On the other hand, it has been suggested that PE is more effective in generating MVPA than other periods (e.g., exergaming, recess, and lunch break) [61] over the school day, in addition to after-school periods that can ensure around 30% of daily MVPA [62]. Furthermore, significant differences between PA levels (total, light, moderate, or vigorous) among weight groups were not found in this study, while a study in Lithuania showed that overweight and obese children had lower PA and participated in MVPA 22.4 min less in comparison to the normal weight group [13].

Total energy costs for structured PA in different weight groups ranged from 1451 kcal/week in UW children not reaching WHO recommendations, to 3081/week kcal in OB children reaching WHO recommendations. As total energy costs depend both on PA levels and weight determined BMR, there are significant differences between kcal used in the same weight group but with different MVPA levels, while we found no difference between the most distinct weight groups for different PA levels. For example, the total energy cost for UW pupils reaching WHO recommendations (2041 kcal/week) was similar to that of OB pupils not reaching the limit (2125 kcal/week). Children who used more kcal during PA also chose bigger portions of carbohydrate-containing foods and bigger portions of meat or fish, irrespectively of the weight group. At the same time, boys, besides significantly higher PA, also put more carbohydrate and protein rich foods on their plate and tended to eat more calorie dense snacks, which could add to previous explanations on obesity prevalence in girls and boys in this study. Similar results were found in the second wave of the population-based German Health Interview and Examination Survey for Children and Adolescents (2014–2017) [63]. On the other hand, current data shows the relation between increased PA and lower number of unhealthy EHs. These findings are in line with a previously mentioned study reporting higher PA level association with higher fruit/vegetable and lower soft drink consumption [63]. On the other hand, relations among any EH and PA in 7–10-year-old Italian children were not found [64]. Finally, the relation between enjoyment of PA and lower BMI and WC was found, pointing to the importance of facilitating pleasant PA in early childhood.

### 4.3. Impact of PA and EH on BMI

Despite the significant differences observed in various factors measured among different weight groups, the present study data failed to explain the variance in BMI, while only the number of vegetable and fruit portions per day was a significant predictor, of small effect size, for WC. Previously published data suggest that obesity is primarily determined by inherited genetic factors explaining 30%–70% variance in BMI [65,66]. Diet-related modifiable factors affecting childhood obesity include nutrient rich but low calorie foods, as well as eating habits described in this study [67]. A recent large-scale population study involving more than 250,000 children and adolescents suggested that current behaviors like television viewing and frequent nut consumption are more important predictors for BMI than early exposure to antibiotics, for example [68]. In addition, it has been shown that nutrition-related parenting practices were associated with BMI in children [69]. Finally, PA has been shown as a factor influencing BMI [70]. As reported previously, EH and PA have a comparatively small effect size on anthropometric measurements and thus a larger sample size as well as the inclusion of other factors (e.g., socio-economic status and parenting practices) could possibly have explained BMI and WC variance to a greater extent. 

## 5. Limitations

One of the possible limitations was that children could under- or over-report their actual EH and PA levels. However, in the current study we found objective differences within weight groups and sex. Another limitation could be that some pupils from the total sample size did not correctly understand the meaning of the question in the questionnaire due to inability to read it properly or maintain focus. To reduce this limitation, electronic questionnaires were designed to be as short as possible and illustrated with pictures. In addition, a researcher or a schoolteacher was available for any questions while children filled out the questionnaires. Slight deviations in the collection of anthropometric measurements by school nurses could be present, despite the provided instructions. This paper represents the baseline cross-sectional data of a longitudinal, prospective study, so any direction of association between the variables cannot yet be determined. Follow up data will be collected to confirm how the change in body mass index and waist circumference is related to dietary and physical activity variables.

## 6. Conclusions

A healthy lifestyle consists of a cluster of different habits and includes a proper diet and PA. EH and PA levels differ between sexes. Girls make healthier dietary choices (e.g., consume more fruits and vegetables, drink more water, and eat smaller portions of carbohydrate and protein rich foods), while boys spend more time in MVPA. In line with other studies, current data describe lower fruit and vegetable consumption among obese children. In addition, obese children skip breakfast more often. Furthermore, children who choose to have their meals in front of TV have higher BMI and WC, which in turn could be explained by a higher number of unhealthy EHs, for example, inadequately low consumption of vegetables and high consumption of calorie dense snacks. Additional structured PA in school (5 vs 2 PE lessons) and other training on weekdays cover weekly WHO recommendations for MVPA (420 min) in less than 20% of pupils. Association between factors (EH and time spent in PA) and BMI was not found in this study. 

## Figures and Tables

**Figure 1 nutrients-12-03818-f001:**
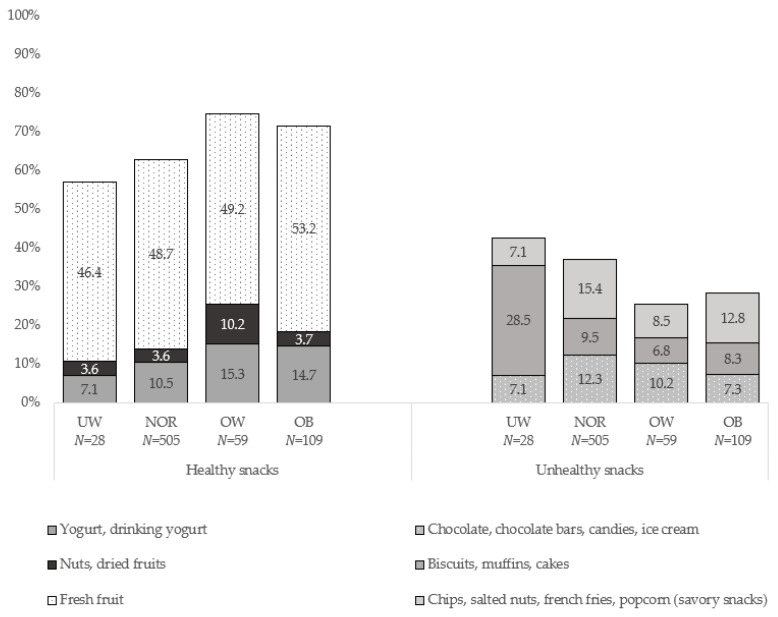
Proportion of snack intake in various weight groups. UW—underweight; NOR—normal weight; OW—overweight; OB—obese.

**Table 1 nutrients-12-03818-t001:** General anthropometric characteristics of the study groups and by sex.

	UW (*N* = 81)	NOR (*N* = 1190)	OW (*N* = 155)	OB(*N* = 219)	Boys (*N* = 890)	Girls (*N* = 759)
Weight, kg	24 (22;26)	29 (26;32)	36 (33;39)	43 (39;50)	31 (28;35)	30 (26;34) *
Waist, cm	55 (53;58)	59 (56;61)	66 (62;70)	76 (70;81)	60 (57;65)	59 (55;64) *
BMI, kg/m^2^	13.5 (13;14)	16 (15;17)	19 (19;20)	22 (21;24)	17 (15;18)	16 (15;18) *

UW—underweight; NOR—normal weight; OW—overweight; OB—obese. * Significant difference compared to boys (*p* < 0.001).

**Table 2 nutrients-12-03818-t002:** Eating habits among the study groups and by sex.

	UW	NOR	OW	OB	Boys	Girls
Eating breakfast every day (% of pupils)						
% (N)	66.7 (52)	60.0 (677)	57.6 (83)	50 (102) *	59.8 (536)	58.6 (465)
Fruit and vegetable consumption (number of portions per day)						
Median (Q1;Q3) N	4 (3;5) 78	4 (3;5) 1128	4 (3;6) * 144	4 (3;5) 203	4 (3;5) 896	4 (3;5) 794
Fruit and vegetable consumption (% of pupils eating recommended ≥ 5 portions per day)						
% (N)	43.6 (34)	46.4 (523)	49.3 (71)	38.2 (78)	43.2 (387)	49.0 (389) **
Vegetable consumption (% of pupils eating recommended ≥ 3 portions per day)						
% (N)	30.8 (24)	37.1 (418)	41.7 (60)	28.4 (58) *	34.4 (308)	38.2 (303)
Fruit consumption (% of pupils eating recommended ≥ 2 portions per day)						
% (N)	65.4 (51)	73.0 (824)	78.5 (113)	66.7 (136) *	71.9 (644)	73.7 (585)
Choice of beverage (% of pupils choosing water)						
%	39.7 (31)	41.0 (463)	36.1 (52)	45.6 (93)	35.5 (318)	48.1 (382) **
Eating culture (% of pupils eating in front of TV or while scrolling the telephone)						
%	26.9 (21)	27.5 (310)	34.0 (49)	35.3 (72)	32.8 (294)	24.3 (193) **
Number of total unhealthy eating habits						
Median (Q1;Q3) N	2 (1;3) 28	2 (1;3) 505	2 (0;3) 59	2 (1;3) 109	2 (1;3) 409	1 (0;3) ** 345

UW—underweight; NOR—normal weight; OW—overweight; OB—obese. TV—television. * Significant difference compared to other weight groups (*p* < 0.05). ** Significant difference compared to boys (*p* < 0.05).

**Table 3 nutrients-12-03818-t003:** Anthropometric parameters and eating habits in pupils eating at the table or in front of the screen.

	BMI, kg/m^2^	Waist, cm	Vegetables, Portions per Day	Calorie Dense Snack, Ranks	Number of Unhealthy Eating Habits (EH)
Pupils Eating at the Table					
Median (Q1; Q3) N	16 (15;18) 1102	60 (56;64) 1070	2 (2;3) 1203	1 (1;3) 546	1 (0;2) 546
Pupils Eating in Front of the Screen					
Median (Q1; Q3) N	17 (15;19) * 452	61 (57;66) ** 443	2 (1;3) ** 487	3 (1;4) ** 208	3 (2;4) ** 208

* Significant difference between groups (*p* < 0.05), ** Significant difference between groups (*p* < 0.001).

**Table 4 nutrients-12-03818-t004:** Meal portion size by sex.

	Boys (*N* = 896)	Girls (*N* = 794)
Carbohydrate Rich Food, Grams per Portion Mean Ranks	253 (175; 331) 903	253 (175; 331) * 781
Vegetable/Salad, Grams per Portion Mean Ranks	85 (56; 145) 843	85 (56; 145) 848
Protein Rich Food, Grams per Portion Mean Ranks	168 (108; 191) 908	108 (56; 168) * 775

* Significant difference compared to boys (*p* < 0.05).

**Table 5 nutrients-12-03818-t005:** Physical activity levels by sex.

	Boys (*N* = 896)	Girls (*N* = 794)
Total PA, min	550 (375;650)	550 (350;625) *
Light PA, min	200 (110;295)	265 (200;295) *
Moderate, min	295 (190;390)	255 (175;350) *
Moderate-to-vigorous physical activity (MVPA), min	330 (210;390)	270 (175;370) *
Prevalence (%) of pupils reaching the recommended daily amount of MVPA during school time and trainings per 5 days	57.9	40.7 **

* Significant difference compared to boys (*p* < 0.05). ** Significant difference compared to boys (*p* < 0.001).

**Table 6 nutrients-12-03818-t006:** Total energy costs of the study groups and by sex.

UW	NOR	OW	OB	Boys	Girls
Total Energy Cost (kcal/week) of Pupils Used for Structured Physical Activities Lasting ≥ 300 min/week					
2011 (1704;2284) *^,#^ *N* = 30	2314 (2039;2663) *N* = 502	2643 (2335;3058) *N* = 59	2991 (2613;3441) *N* = 77	2580 (2282;2971) *N* = 410	2113 (1865;2377) *N* = 259
Total Energy Cost (kcal/week) of Pupils Used for Structured Physical Activities Lasting < 300 min/week					
1527 (1268;1709) * *N* = 18	2314 (1419;1804) * *N* = 269	1792 (1667;1922) *N* = 44	2070 (1852;2329) *N* = 55	1807 (1571;2005) *N* = 175	1567 (1390;1759) *N* = 211

UW—underweight; NOR—normal weight; OW—overweight; OB—obese. No significant difference compared to NOR pupils with structured PA lasting < 300 min/week (*p* > 0.05). * No significant difference compared to OW pupils with structured PA lasting < 300 min/week (*p* > 0.05). ^#^ No significant difference compared to OB pupils with structured PA lasting < 300 min/week (*p* > 0.05).

**Table 7 nutrients-12-03818-t007:** Correlation coefficients characterizing relations between independent (measures) and dependent (anthropometric measurements) variables.

	BMI, kg/m^2^	Waist, cm
Number of Daily Meals, Meals per Day	*r* = 0.019	*r* = −0.003
Breakfasts Frequency	*r* = −0.082 ***	*r* = −0.100 ***
Carbohydrate Portion, gr	*r* = 0.030	*r* = 0.043
Meat/Fish Portion, gr	*r* = 0.021	*r* = 0.045
Vegetable Portion, gr	*r* = −0.017	*r* = 0.007
Vegetables and Fruit, Portions per Day	*r* = −0.035	*r* = −0.044 *
Calorie Dense Snacks	*r* = −0.063 *	*r* = −0.059 *
Enjoying Physical Activities	*r* = −0.051 **	*r* = −0.079 **
Time Spent in MVPA, min	*r* = 0.012	*r* = −0.004
Additional Trainings, Number per Week	*r* = −0.006	*r* = −0.055 *

* Tendency (*p* < 0.1). ** Significant difference (*p* < 0.05). *** Significant difference (*p* < 0.001).

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
