# Peer review of "Comparative Analysis of the Effects of Daily Eating Habits and Physical Activity on Anthropometric Parameters in Elementary School Children in Latvia: Pach Study"

_nutrients, 2020, doi:10.3390/nu12123818_

Round 1

Reviewer 1 Report

Childhood obesity is a global problem. However, the manuscript is not structured as per the format for a research journal. The title, design, purpose, measures, analysis, and results need to be linked so that readers can understand the study.

1. Title and Purpose

You did not analyze the data using multiple regression. Readers can mistake EA and PA as significant variables that affect anthropometric parameters considering demographics and other variables.

You should link the research title with the research purpose.

You need to describe the research objectives in detail. 

2. Abstract

Lines 27-28: Did the screening time result in higher BMI and WC? How do you define a causal relationship? You did not analyze the data using multiple regression.

3. Study design

What is the study design?

Your methodology does not have an experimental design. Line 124: Why is the experimental design described?

4. Measures

According to the research purpose and analysis method, you should describe the dependent variables and the outcome variables and measures.

Lines 71-72: You described variables are related to obesity for the purpose of this study.

Line 266: You described that you used a multiple regression model. What is the dependent variable in the model?

Line 100-123: Why was the total energy cost described in the questionnaires section?

Line 124-138: Why are measurement mentioned in the experimental design section?

Line 33: How did you measure screen time?

Line 110: You should summarize and describe in the main text the variables from the appendix.

5. Procedures

You should describe study procedures in the procedure section.

You should describe what data were collected, by whom, when, and in what way.

6. Statistical analysis

Why are the BMI classifications mentioned in the statistical analysis? They should be described in the measures section.

Line 266: The multiple regression model you used should be described in the statistical analysis section.

7. Results 

Subheadings should be described according to detailed research objectives.

8. Tables and figures 

You should provide PA level according to the anthropometric parameter group in Table 2.

You should provide statistical values in in the tables (t, F, z, etc.).

Figure 1 is not informative.

If you want to suggest the level of exercise with respect to gender, you should describe it in research objectives.

9. English

English tense is not correct. For example, lines 88, 91: “describes”, “involves”

Except for the abstract, the full term for every abbreviation should be described upon first use. The same is true for abbreviations of figures or tables.

Author Response

Dear reviewers,

We thank you all for your valuable suggestions that have helped to improve our manuscript.

Now we have made major revisions which include:

  • More clear Title and Objectives of the study.
  • Abstract without misleading causal relations.
  • Completely reorganized Materials and Methods, which now include subsections Study design and subjects, Measures (describing all the dependent and codependent variables more in detail), Procedure and Statistical analysis.
  • Results section now provide a lot more data in tables describing differences among study groups and sex, relations between measured parameters and multiple linear regression analysis. Subsections of results are in line with Measures described in Materials and Methods as well as with study objectives.
  • Additional literature is provided in discussion part.
  • Conclusions have been rewritten to show more clearly their justification based on results gained in this study.

We hope that the manuscript now is structured as per the format for a research journal and that all of the parts are integrated in a way that the readers can understand the study. Once more thank you for your contribution in the preparation of this manuscript.

In a name of whole research team,

Project leader

Dr. biol. Dace Reihmane

Reviewer 1

Yes

Can be improved

Must be improved

Not applicable

Does the introduction provide sufficient background and include all relevant references?

( )

( )

(x)

( )

Is the research design appropriate?

( )

( )

(x)

( )

Are the methods adequately described?

( )

( )

(x)

( )

Are the results clearly presented?

( )

( )

(x)

( )

Are the conclusions supported by the results?

( )

(x)

( )

( )

Comments and Suggestions for Authors

Childhood obesity is a global problem. However, the manuscript is not structured as per the format for a research journal. The title, design, purpose, measures, analysis, and results need to be linked so that readers can understand the study.

A: In a name of whole research team I would like to thank you for the valuable suggestions you have given that helped to perform major revision of the manuscript. We hope that improvements made will be sufficient for publishing in Nutrients.

  1. Title and Purpose

You did not analyze the data using multiple regression. Readers can mistake EA and PA as significant variables that affect anthropometric parameters considering demographics and other variables.

You should link the research title with the research purpose.

You need to describe the research objectives in detail. 

A: We have performed multiple regression analysis; however, model was not significant. Thus we have now adapted the study title and objectives in order to bring more clarity and structure for readers.

  1. Abstract

Lines 27-28: Did the screening time result in higher BMI and WC? How do you define a causal relationship? You did not analyze the data using multiple regression.

A: We agree with your remark, now we have clarified that “pupils having meals in front of screen have unhealthier EH and anthropometric measures” without misleading for causal relationship.

  1. Study design. What is the study design? Your methodology does not have an experimental design. Line 124: Why is the experimental design described?

A: Thank you for the remark, we have now rearranged the section of methods and changed the title to “Study design and subjects”. We also have added extra information in this subsection.

  1. Measures

According to the research purpose and analysis method, you should describe the dependent variables and the outcome variables and measures.

Lines 71-72: You described variables are related to obesity for the purpose of this study.

Line 266: You described that you used a multiple regression model. What is the dependent variable in the model?

Line 100-123: Why was the total energy cost described in the questionnaires section?

Line 124-138: Why are measurement mentioned in the experimental design section?

Line 33: How did you measure screen time?

Line 110: You should summarize and describe in the main text the variables from the appendix.

A: Thank you for clarifying importance of measures section from reader’s perspective. This section now includes 5 subsections (Anthropometric parameters, Eating habits, Meal portion size, Time spent in structured physical activities, Total energy costs during structured physical activities) which describes all the dependent and codependent variables more in detail. Please see more details in manuscript.

  1. Procedures

You should describe study procedures in the procedure section.

You should describe what data were collected, by whom, when, and in what way.

A: We have taken your suggestion into consideration and left in section Procedure only the relevant information.

  1. Statistical analysis

Why are the BMI classifications mentioned in the statistical analysis? They should be described in the measures section.

Line 266: The multiple regression model you used should be described in the statistical analysis section.

A: Thank you for your comment. We have now added information on analysis methods, while transferred BMI classifications to section “Measures”.

  1. Results 

Subheadings should be described according to detailed research objectives.

A: We looked at several already published manuscripts, however, couldn’t find detailed research objectives. However, results now include 7 subsections, 5 of them in the same order as described measures in section Materials and methods and 2 additional ones describing data on performed correlations and multiple regression analysis. We hope this answers your concerns about subheadings.

  1. Tables and figures 

You should provide PA level according to the anthropometric parameter group in Table 2.

You should provide statistical values in in the tables (t, F, z, etc.).

Figure 1 is not informative.

If you want to suggest the level of exercise with respect to gender, you should describe it in research objectives.

A: We have now included a lot more data in form of tables describing both differences among weight groups as well as sex, providing also corresponding statistical values. We hope that will bring more clarity for readers.

  1. English

English tense is not correct. For example, lines 88, 91: “describes”, “involves”

Except for the abstract, the full term for every abbreviation should be described upon first use. The same is true for abbreviations of figures or tables.

A: Thank you for your remarks. We have once more read the text and have payed special attention to the tense used in the text. There is no information in author guidelines for use of tense. We looked at several previously published manuscripts in this journal and concluded that past tense is used while describing the study data, while present tense for drawn general conclusions. We have now followed this principle also in our manuscript.

Reviewer 2 Report

This is a well-conducted prevalence study.
The summary and introduction are appropriate.
The methodology could be better explained.
I suggest changing the title "experimental design" on page 3 to research
design since it is not an experimental study (variables are not manipulated)
and it may confuse the reader.
Other suggestion is to
explain more deeply the results of multivariate
regression
since it seems to me one of the main contributions of this study.
Why do they say that there is no clinical relevance?
Authors
should present the regression equation even if the beta coefficients
are not significant and discuss the possible reasons for which it has happened.

Finally, it is recommended to include future lines of research.

Author Response

Dear reviewers,

We thank you all for your valuable suggestions that have helped to improve our manuscript.

Now we have made major revisions which include:

  • More clear Title and Objectives of the study.
  • Abstract without misleading causal relations.
  • Completely reorganized Materials and Methods, which now include subsections Study design and subjects, Measures (describing all the dependent and codependent variables more in detail), Procedure and Statistical analysis.
  • Results section now provide a lot more data in tables describing differences among study groups and sex, relations between measured parameters and multiple linear regression analysis. Subsections of results are in line with Measures described in Materials and Methods as well as with study objectives.
  • Additional literature is provided in discussion part.
  • Conclusions have been rewritten to show more clearly their justification based on results gained in this study.

We hope that the manuscript now is structured as per the format for a research journal and that all of the parts are integrated in a way that the readers can understand the study. Once more thank you for your contribution in the preparation of this manuscript.

In a name of whole research team,

Project leader

Dr. biol. Dace Reihmane

Reviewer 2

Yes

Can be improved

Must be improved

Not applicable

Does the introduction provide sufficient background and include all relevant references?

(x)

( )

( )

( )

Is the research design appropriate?

( )

(x)

( )

( )

Are the methods adequately described?

(x)

( )

( )

( )

Are the results clearly presented?

( )

(x)

( )

( )

Are the conclusions supported by the results?

(x)

( )

( )

( )

Comments and Suggestions for Authors in italic and provided answers (A)

This is a well-conducted prevalence study. The summary and introduction are appropriate.

A: Thank you for the comment, some major adjustments though has been made as asked by the other reviewers.

The methodology could be better explained. It is necessary to delve deeper into the type of study (quantitative, by survey ...) and eliminate the title "experimental design", they couldc hange it by other more appropriate, for example, "procedure". I suggest changing the title "experimental design" on page 3 to research design since it is not an experimental study (variables are not manipulated) and it may confuse the reader.

A: We have followed your justified suggestion and changed the subtitle as well as modre in detailed described the measures.

Other suggestion is to explain more deeply the results of multivariate regression since it seems to me one of the main contributions of this study. Why do they say that there is no clinical relevance? Authors should present the regression equation even if the beta coefficients are not significant and discuss the possible reasons for which it has happened.

A: Thank you for your suggestion. Now we have made even more detailed multi linear regression analysis (8 predictors) for dependent variables BMI and WC. However, the models were not significant. To some extent results could be explained by study limitations.

Finally, it is recommended to include future lines of research.

A: Thank you for remark. We have now includes brief final paragraph in section “Discussion” describing future research directions in this cohort, lines 448-450.

Reviewer 3 Report

A greatest advantage of the publication is a large group of respondents, but the theses presented in the study unfortunatelly do not bring anything new to the current state of knowledge in this area

Detailed comments:

Is a description of how gender was determined necessary?

On line 117, give the full name of the acronym( METy), it  appears for the first time.

On what basis was the division into two groups of children with good and bad eating habits, why> 3 and<3 and not the mean value and SD for the studied group?

The term “irregular meals” was used, but was it not better to use the term “ the number of meals during the day”? Irregular meals this term I more about the intervals between each meal, not the number of meals. (line 163)

In line 263 there is a lack of information about the force of the interaction between the traits.

In the abstract following sentences are written  “Increasing MVPA is related to smaller  number of unhealthy EH Higher levels of MVPA are associated with healthier EH, while screen time during meal increase  unhealthy EH, resulting in higher BMI and WC. Structured PA on weekdays is not sufficient for  optimal children health.”  My question is : where the results for these conclusions are presented?

There are no results for the comparison of the relation between nutritional behavior, anthropometric parameters and physical activity, these results are crucial and should be shown  in table to better present the relation.

The results for following statement should be presented it table: Higher levels of MVPA are associated with healthier EH, while screen time during meal increase unhealthy EH, resulting in higher BMI and WC.

Definitely too few results are presented in the tables, basically limited to showing only percentage values in the population for individual variables without showing differences between the variables that are the subject of the work (eating behavior, physical activity and BMI).

In the discussion section in the part of  fruit and vegetables, there is no discussion about the impact of the increased  fruit and vegetable consumption on anthropometric parameters

In the discussion section in the part on physical activity, there are no studies that show the influence of physical activity on anthropometric parameters

The observations presented as novelity in the abstract are in fact not new, many previous studies have confirmed the relation between physical activity, eating behavior and BMI in children.

Author Response

Dear reviewers,

We thank you all for your valuable suggestions that have helped to improve our manuscript.

Now we have made major revisions which include:

  • More clear Title and Objectives of the study.
  • Abstract without misleading causal relations.
  • Completely reorganized Materials and Methods, which now include subsections Study design and subjects, Measures (describing all the dependent and codependent variables more in detail), Procedure and Statistical analysis.
  • Results section now provide a lot more data in tables describing differences among study groups and sex, relations between measured parameters and multiple linear regression analysis. Subsections of results are in line with Measures described in Materials and Methods as well as with study objectives.
  • Additional literature is provided in discussion part.
  • Conclusions have been rewritten to show more clearly their justification based on results gained in this study.

We hope that the manuscript now is structured as per the format for a research journal and that all of the parts are integrated in a way that the readers can understand the study. Once more thank you for your contribution in the preparation of this manuscript.

In a name of whole research team,

Project leader

Dr. biol. Dace Reihmane

Reviewer 3

Yes

Can be improved

Must be improved

Not applicable

Does the introduction provide sufficient background and include all relevant references?

( )

( )

(x)

( )

Is the research design appropriate?

( )

(x)

( )

( )

Are the methods adequately described?

( )

(x)

( )

( )

Are the results clearly presented?

( )

( )

(x)

( )

Are the conclusions supported by the results?

( )

( )

(x)

( )

Comments and Suggestions for Authors in italic and provided answers (A)

A greatest advantage of the publication is a large group of respondents, but the theses presented in the study unfortunately do not bring anything new to the current state of knowledge in this area.

Detailed comments:

Is a description of how gender was determined necessary?

A: Thank you for the question. In some journals recently it has been as a requirement, however, not for Nutrients, so we have taken out the description.

On line 117, give the full name of the acronym (METy), it appears for the first time.

A: Thank you for the remark. Full name of the acronym (METy) now has been provided. In text mentioned – “youth metabolic equivalents”.

On what basis was the division into two groups of children with good and bad eating habits, why> 3 and<3 and not the mean value and SD for the studied group?

A: After major revisions done based on your and other reviewers valuable comments for data representation, we have now excluded this scoring system from the study as it did not have the added value in further analysis.

The term “irregular meals” was used, but was it not better to use the term “the number of meals during the day”? Irregular meals this term I more about the intervals between each meal, not the number of meals. (line 163)

A: Yes, we agree with the objection. The manuscript has been clarified at this point. The irregularity/regularity studied in the study relates to the frequency of meal, namely, the number of meals per day.

In line 263 there is a lack of information about the force of the interaction between the traits.

A: Thank you for the remark. Correlation coefficient has been added.

In the abstract following sentences are written “Increasing MVPA is related to smaller number of unhealthy EH Higher levels of MVPA are associated with healthier EH, while screen time during meal increase unhealthy EH, resulting in higher BMI and WC. Structured PA on weekdays is not sufficient for optimal children health.”  My question is: where the results for these conclusions are presented?

A: We have now written abstract in a way that conclusions can be drawn from data presented in abstract. We hope that rearrangements made will facilitate readers understanding.

There are no results for the comparison of the relation between nutritional behaviour, anthropometric parameters and physical activity, these results are crucial and should be shown in table to better present the relation. The results for following statement should be presented it table: Higher levels of MVPA are associated with healthier EH, while screen time during meal increase unhealthy EH, resulting in higher BMI and WC. Definitely too few results are presented in the tables, basically limited to showing only percentage values in the population for individual variables without showing differences between the variables that are the subject of the work (eating behaviour, physical activity and BMI).

A: We agree to your opinion. Now paper contains in total 7 tables describing the study data. Additional section in results named “3.6. Correlations between measured parameters” which more in detail explains the relations between various measured factors (Table 7). In addition, Table 3 now provides data between pupils eating in front of the screen or at the table. We hope it improves the understanding of the study results.

In the discussion section in the part of fruit and vegetables, there is no discussion about the impact of the increased fruit and vegetable consumption on anthropometric parameters.

A: We added a paragraph between lines 358-360 with discussion about increased fruit and vegetable consumption effect on anthropometric parameters.

In the discussion section in the part on physical activity, there are no studies that show the influence of physical activity on anthropometric parameters.

A: Thank you for suggestion. In line 425-428 we added comparison of our findings with our colleague study from the neighbour country Lithuania.

The observations presented as novelty in the abstract are in fact not new, many previous studies have confirmed the relation between physical activity, eating behaviour and BMI in children.

A: Thank you for the comment. You are right, it is not novelty for worldwide research community though it is confirmation for local stakeholders. Now we have removed paragraph Novelty from the manuscript.

Round 2

Reviewer 1 Report

Through the previous revision, the title, design, purpose, measures, procedures, analysis, and results were linked. The number of tables is high, and they are very lengthy. I understand that the tables were added according to the reviewers’ suggestions; however, the authors should select only those tables that they believe are crucial as regards the study’s aim. In particular, scientific tables should be well-organized, aid the reader’s understanding, and speed up the comprehension of the study’s findings. For example, in the correlation table, why are p values listed in all columns? (Table 7). Another example, the BMI group and sex are listed in the same row (Tables 1 and 2). It should be analyzed in groups according to sex. What do the two rows mean? (Table 3). Additionally, is there a need of mentioning the mean and median in all the tables? I recommend that you simplify the tables by providing the mean & SD for a parametric statistical method and median & Q for a nonparametric method. The tables should be organized more concisely.

In the previous revision, statistical values were suggested. However, in most univariate analyses, only p values were presented (in both, the bottom of the table and text). Please describe the statistical values in the main text (F, t, or Z, etc.).

Please state the total number of participants as (N =     ) in the title of each table.

Explain why EA and PA are not factors that affect anthropometric parameters in multiple regression analysis. Crucial covariates, such as hormonal or familial factors, were not investigated.

Author Response

Dear reviewers,

We would like to thank you for your valuable and useful suggestions. Our research team took them into consideration and according to your review, we have made the following revisions to the manuscript:

  • Data representation in the text was improved by the addition of statistical values.
  • Tables in the manuscript were standardized.
  • Paragraph on the results of multilinear regression analysis was added and additionally discussed in Discussion section.

We hope that our improvements make the manuscript more well-structured and are appropriate to your reviews.

In a name of all research team,

Project coordinator,

Jelena Raudeniece

Reviewer 3 Report

Thank you for making changes to the manuscript, it's way better, I still have comments on the tables. In the tables, some rows contain the p value, in others it is not given, I suggest standardizing the tables in terms of editing.

Author Response

(The authors gave the same response as above.)
